# Identification of Regulatory RNA-Binding Genes in Spermatogonial Stem Cell Reprogramming to ES-like Cells Using Machine Learning–Integrated Transcriptomic and Network Analysis

**DOI:** 10.3390/cells14201632

**Published:** 2025-10-20

**Authors:** Ali Shakeri Abroudi, Hossein Azizi, Hewa Khalid Abdullah, Marwa Fadhil Alsaffar, Thomas Skutella

**Affiliations:** 1Department of Cellular and Molecular Biology, Faculty of Advanced Science and Technology, Tehran Medical Sciences, Islamic Azad University, Tehran 4818986557, Iran; alishakeriabroudi@gmail.com; 2Faculty of Biotechnology, Amol University of Special Modern Technologies, Amol 4616686767, Iran; 3Department of Surgery, Faculty of General of Medicine, Koya University, Koya KOY45 5462844952, Iraq; hewa.abdullah@koyauniversity.org; 4Medical Laboratories Techniques Department, College of Health & Medical Techniques, AL-Mustaqbal University, Hillah 51001, Iraq; marwaalsaffar@uomus.edu.iq; 5Institute for Anatomy and Cell Biology, Medical Faculty, University of Heidelberg, Im Neuenheimer Feld 307, 69120 Heidelberg, Germany; thomas.skutella@uni-heidelberg.de

**Keywords:** spermatogonial stem cells, ES-like cell, single-cell RNA sequencing, reprogramming, germline

## Abstract

Spermatogonial stem cells (SSCs) are unipotent germline cells with emerging pluripotent potential under specific in vitro conditions. Understanding their capacity for reprogramming and the molecular mechanisms involved offers valuable insights into regenerative medicine and fertility preservation. SSCs were isolated from Oct4-GFP C57BL/6 transgenic mice using enzymatic digestion and cultured in defined media. Under these conditions, ES-like colonies emerged expressing pluripotency markers. These cells were characterized by immunocytochemistry, teratoma assays, and transcriptomic analyses using bulk and single-cell RNA sequencing datasets. Gene expression profiles were compared with ESCs and SSCs using datasets from GEO (GSE43850, GSE38776, GSE149512). Protein–protein interaction (PPI) networks and co-expression modules were explored through STRING, Cytoscape, and WGCNA. ES-like cells derived from SSCs exhibited strong expression of OCT4, DAZL, and VASA. Transcriptomic analysis revealed key differentially expressed genes and shared regulatory networks with ESCs. WGCNA identified key co-expression modules and hub regulatory RNA binding genes (Ctdsp1, Rest, and Stra8) potentially responsible for the reprogramming process. Teratoma assays confirmed pluripotency, and single-cell RNA-seq validated expression of critical markers in cultured SSCs. This study demonstrates that SSCs can acquire pluripotency features and be reprogrammed into ES-like cells. The integration of transcriptomic and network-based analyses reveals novel insights into the molecular drivers of SSC reprogramming, highlighting their potential utility in stem cell-based therapies and male fertility preservation.

## 1. Introduction:

Spermatogonial stem cells (SSCs) are a rare and vital subpopulation of germline stem cells residing within the seminiferous tubules of the testis [1]. They are responsible for sustaining spermatogenesis through a delicate balance of self-renewal and differentiation into mature spermatozoa throughout a male’s reproductive lifespan. In addition to their well-established role in gametogenesis, SSCs have garnered significant interest for their developmental plasticity and potential as an alternative source of pluripotent stem cells [2].

Recent studies have demonstrated that under specific in vitro culture conditions, SSCs derived from neonatal or adult testes can dedifferentiate into embryonic stem cell-like (ES-like) cells without the need for exogenous transcription factors [3]. This spontaneous reprogramming process has been shown to result in cells expressing key pluripotency markers such as OCT4, NANOG, and SOX2, as well as the ability to form teratomas and contribute to chimeric embryos [4,5]. Unlike induced pluripotent stem cells (iPSCs), SSC-derived ES-like cells avoid concerns regarding oncogene transduction or epigenetic memory from somatic origins, presenting a safer and more natural route to pluripotency [6].

In mouse models, transgenic approaches have been employed to trace and monitor pluripotency induction, such as the use of OCT4-GFP reporter systems that enable the visual tracking of reprogramming efficiency and stem cell maintenance [7,8]. Culturing SSCs in germline stem cell (mGSC) media enriched with growth factors like GDNF, FGF2, and LIF has been shown to support both their proliferation and transition to a pluripotent state [9,10]. Moreover, the plasticity of SSCs has been corroborated by studies in human testicular cells, indicating conserved molecular mechanisms across species [11].

However, the underlying transcriptional programs and gene regulatory networks that govern the transition from unipotent SSCs to pluripotent ES-like cells remain incompletely understood. High-throughput techniques such as microarrays and single-cell RNA sequencing (scRNA-seq) have begun to reveal the dynamic gene expression changes and lineage trajectories involved in this process [12,13,14]. For instance, recent scRNA-seq analyses have identified previously unrecognized subpopulations of undifferentiated spermatogonia in both mouse and human testis, characterized by novel markers such as ID4, UTF1, and EOMES [15,16].

To address the complexity of gene interactions during SSC reprogramming, systems biology tools have emerged as powerful frameworks. Weighted Gene Co-expression Network Analysis (WGCNA), for example, enables the identification of gene modules that are highly correlated with cellular phenotypes or clinical traits [17]. In parallel, protein–protein interaction (PPI) networks derived from resources such as STRING can elucidate hub genes and key signaling pathways that orchestrate cellular transitions [18]. These integrative approaches provide a more comprehensive understanding of stem cell behavior and hold potential for uncovering therapeutic targets or improving culture systems for regenerative applications.

In this study, we isolated SSCs from OCT4-GFP transgenic C57BL/6 mice and successfully reprogrammed them into ES-like cells using defined culture conditions. We then performed comparative transcriptome profiling using microarray datasets and identified differentially expressed genes (DEGs) between SSCs, ES-like cells, and ESCs. Using STRING and Cytoscape, we constructed interaction networks to reveal central regulators, and applied WGCNA to identify gene modules associated with pluripotency acquisition. Finally, we validated hub genes using publicly available scRNA-seq datasets of human SSCs and quantified gene expression across different cell types using Fluidigm qPCR. Collectively, our findings provide novel insights into the molecular landscape of SSC-derived pluripotency and lay the groundwork for future applications in male fertility preservation and stem cell-based therapies.

In the present work, SSC-derived ES-like cells were generated and evaluated by teratoma and chimera formation assays to confirm their pluripotency potential. These cells were not the primary subject of high-throughput transcriptomic profiling in this study; instead, they provide an experimental proof-of-concept supporting the rationale for subsequent in silico analyses of publicly available datasets (GEO and TCGA). The focus of the bioinformatic analyses was to investigate SSC-associated transcriptional networks in relation to human male infertility. The objective of this study was to identify key regulatory RNA-binding genes and co-expression modules involved in SSC reprogramming into ES-like cells. By integrating transcriptomic and network-based analyses, we aimed to uncover molecular drivers of pluripotency acquisition and provide insights into their potential application in male fertility preservation and regenerative medicine.

## 2. Materials and Methods

### 2.1. Isolation SSCs

The use of animals for spermatogonial stem cell isolation was reviewed and approved by the Institutional Animal Care and Ethics Committee at Amol University of Special Modern Technologies (approval letter No. Ir.asmt.rec.1403.125, issued on 12 May 2025). We used Oct4 promoter-driven GFP reporter C57BL/6 transgenic mice (OCT4-GFP). These mice were generated as previously described by Kim and Lee [5] and Hashemi Karoii et al. [9] and maintained on a C57BL/6 genetic background. The breeding colony was established and maintained in-house at the animal facility of Amol University of Special Modern Technologies under specific-pathogen-free (SPF) conditions. Genotyping was performed according to the published protocol to confirm the presence of the Oct4-GFP transgene before experiments [10,11]. Animal care and ethics at Amol University of Special Modern Technologies were reviewed and authorized for this research. This was accomplished by harvesting testicular cells from Oct4 promoter reporter-equipped C57BL/6 GFP transgenic mice. Before killing with CO2 gas, the mice were gently put to sleep by inducing unconsciousness with a regulated flow rate of around 30–70% of the chamber capacity per minute. The mice were carefully observed to ensure they were asleep before their respiration was stopped. Because it is effective and generally seen as a humane method of killing rats, this method was selected. Additional local anesthetics were not administered. For the purpose of cell separation, a simple enzymatic digestion procedure was used. The testicular tissue was digested for 8 min at 37 °C using an enzymatic solution that included Collagenase IV, Dispase, DNase, and HBSS buffer. The addition of 10% ES cell-qualified FBS stopped the enzymatic digestion after careful pipetting achieved a single-cell suspension. Afterwards, the samples were centrifuged for 10 min at 1500 rpm, washed with a DMEM/F12 combination, filtered using a 70 μm cell strainer, and then subjected to additional centrifugation. For cell separation, testicular tissue was subjected to enzymatic digestion at 37 °C for 8 min using a solution containing 1 mg/mL Collagenase IV (Sigma-Aldrich, Massachusetts, United States, Cat. No. C5138), 1 mg/mL Dispase II (Sigma-Aldrich, Cat. No. D4693), and 0.5 mg/mL DNase I (Sigma-Aldrich, Cat. No. DN25) prepared in Hank’s Balanced Salt Solution (HBSS). The tissue was gently pipetted during digestion to facilitate dissociation into single-cell suspension. Enzymatic activity was stopped by adding 10% ES cell-qualified fetal bovine serum (FBS, Gibco). The suspension was centrifuged at 1500 rpm for 10 min, washed once with DMEM/F12 medium, passed through a 70 µm cell strainer, and subjected to a second centrifugation step before culture.

### 2.2. Culture of SSCs

Following enzymatic digestion and filtration, single-cell suspensions were counted with a hemocytometer and trypan blue. Cells were seeded onto tissue culture plates pre-coated with 0.2% gelatin at 1.0 × 10^5^ viable cells per well of a 6-well plate (≈1.7 × 10^4^ cells/cm^2^). SSCs were maintained in mGSC medium based on StemPro-34 (Thermo Fisher) supplemented to the final concentrations listed below: N2 supplement, 1× (manufacturer’s recommended final dilution); D-glucose, 4.5 g·L^−1^ (high-glucose level, ≈25 mM); bovine serum albumin (BSA), 0.1% *w*/*v*; L-glutamine, 2 mM; α-mercaptoethanol, 0.1 mM; MEM vitamins, 1×; non-essential amino acids (NEAA), 1×; 17β-estradiol, 10 nM; progesterone, 100 nM; Epidermal growth factor (EGF), 20 ng·mL^−1^; basic fibroblast growth factor (bFGF/FGF2), 10 ng·mL^−1^; Glial cell line-derived neurotrophic factor (GDNF), 20 ng·mL^−1^; Leukemia inhibitory factor (LIF), 100 U·mL^−1^; ES cell-qualified fetal bovine serum (FBS), 1% *v*/*v*; ascorbic acid (vitamin C), 50 µg·mL^−1^; sodium pyruvate, 1 mM; and DL-lactic acid, 10 mM. Penicillin/streptomycin was added at 100 U·mL^−1^/100 µg·mL^−1^. Cultures were kept at 37 °C, 5% CO_2_. Medium was partially (~50%) replaced every 48 h, and fully refreshed every 4 days; cells were monitored daily and passaged (mechanically or by gentle trypsinization) at ~70–80% confluence, typically every 7–10 days. All growth factor stocks were prepared according to the manufacturers’ instructions and diluted into medium immediately before use [12].

### 2.3. Generation and Culture of ES-like Cells Derived from SSC

In the past, we generated transgenic mice with a GFP reporter linked to the OCT4 promoter using a mouse spermatogonial stem cell medium. The C57BL/6 strain was used for these mice. Within 41–125 days of initiating the culture, we were successful in producing ES-like cells expressing high quantities of OCT4-GFP. Following that, the cells were isolated and maintained in a mESCs medium containing KO-DMEM (or high-glucose DMEM), 15% fetal bovine serum (FBS), MEM non-essential amino acids (NEAA), L-glutamine, Penicillin-Streptomycin (Pen-Strep), γ-mercaptoethanol, and 1000 U/mL of leukemia inhibitory factor (LIF). To keep them growing, they were transferred to new layers of embryonic fibroblasts in mice every several days. OCT4-GFP transgenic mice were generated by pronuclear microinjection of a construct in which enhanced green fluorescent protein (EGFP) was driven by a 2.4 kb fragment of the murine Oct4 promoter, as previously described [13,14]. Founder mice were screened by PCR, and positive animals were bred to establish stable lines. For this study, animals were obtained from our in-house colony originally derived from the published strain (B6;CBA-Tg(Pou5f1-EGFP)2Mnn/J, The Jackson Laboratory, Stock No. 004654). All animals were maintained under specific pathogen-free conditions, and experiments were performed in accordance with institutional animal care and use guidelines. Emerging ES-like colonies were identified based on characteristic compact, dome-shaped morphology under phase-contrast microscopy. Colonies were manually picked using a finely pulled glass capillary under a stereomicroscope following brief enzymatic dissociation with 0.05% trypsin-EDTA to loosen cell clusters. Individual colonies were transferred to fresh gelatin-coated plates and expanded in mESC medium (DMEM supplemented with 15% FBS, 2 mM L-glutamine, 0.1 mM non-essential amino acids, 0.1 mM β-mercaptoethanol, and 1000 U/mL LIF).

### 2.4. Teratoma Assay

For teratoma formation, ~1 × 10^6^ ES-like cells were resuspended in 50 µL of PBS and injected subcutaneously into the dorsal flank of 6–8-week-old immunodeficient NOD/SCID mice under isoflurane anesthesia. Mice were monitored daily, and tumors were collected 6–8 weeks post-injection, fixed in 4% paraformaldehyde, embedded in paraffin, and sectioned for hematoxylin and eosin staining to identify derivatives of all three germ layers.

For chimera assays, 3–4-week-old female B6D2F1 mice were superovulated by intraperitoneal injection of 5 IU pregnant mare serum gonadotropin (PMSG), followed 48 h later by 5 IU human chorionic gonadotropin (hCG), and then mated with fertile males. Blastocysts were harvested from the uteri at 3.5 days post coitum by flushing with M2 medium (Sigma-Aldrich) and maintained in KSOM medium at 37 °C, 5% CO_2_ until use. Approximately 10–15 ES-like cells were injected into the blastocoel cavity of each blastocyst using a Piezo-driven micromanipulator under an inverted microscope. After injection, 8–10 blastocysts were transferred into each uterine horn of 2.5-day pseudopregnant ICR females under isoflurane anesthesia via a midline laparotomy. The abdominal incision was sutured in two layers, and animals were provided analgesia (meloxicam, 1 mg/kg, s.c.) and monitored until full recovery. Pups were delivered naturally and assessed for chimerism based on coat color contribution and confirmed by PCR genotyping where applicable.

### 2.5. Immunocytochemical Staining

Careful processing of testicular cells was carried out using a 4% paraformaldehyde fixation solution and a 0.1% Triton X-100 solution in PBS permeabilization method. To prevent non-specific binding, a 1% BSA in PBS solution is used as a blocking step. Primary antibodies against OCT4, VASA, and DAZL were used in immunocytochemistry and immunohistochemistry investigations. These antibodies were acquired from different American companies: Bio-Rad (Hercules, CA, USA), Santa Cruz Biotechnology (Dallas, TX, USA), and Abcam (Cambridge, UK), respectively. Following the incubation of the primary antibodies, secondary antibodies (goat anti-mouse IgG H&L, Abcam, Cambridge, UK) were used. To stain the cells, we used 0.2 g/mL of 4′,6-diamidino-2-phenylindole (DAPI) and left them at room temperature for three minutes. Then, in order to see the cell interiors, they were glued with mowiol, a polyvinyl alcohol. Pictured here are positively tagged cells captured using a confocal Zeiss LSM 700 microscope (Carl Zeiss, Oberkochen, Germany). The images were captured by use of a Zeiss LSM-TPMT lens.

### 2.6. Microarray Data Analysis

Using data available from the NCBI Gene Expression Omnibus (https://www.ncbi.nlm.nih.gov/geo/ Date: 10 August 2024), we conducted microarray analysis using ES-like cell samples and ESC samples from the GSE43850 [15] dataset, and SSC expression data from the GSE38776 dataset. For transcriptomic profiling, publicly available datasets were obtained from the NCBI Gene Expression Omnibus (GEO). Specifically, 3 embryonic stem cell (ESC) samples and 3 ES-like cell samples were included from dataset GSE43850, and 3 spermatogonial stem cell (SSC) samples were included from dataset GSE38776.

Since the datasets were generated on different array platforms, all raw CEL files were reprocessed to ensure comparability. Data normalization was performed using the Robust Multi-array Average (RMA) method in the Transcriptome Analysis Console (TAC v4.0). To correct for potential batch effects between platforms, we applied the ComBat function from the sva package in R, which preserves biological variation while removing technical biases.

After normalization and batch correction, a combined expression matrix was generated. Differential gene expression analysis was then carried out using the empirical Bayes (eBayes) ANOVA method. Genes with a log2 fold change ≥ |2| and adjusted *p* < 0.05 were considered significant DEGs for stringent analysis. In addition, a relaxed threshold of log2 fold change ≥ |2| was also applied in secondary analyses to capture additional biologically relevant candidates for downstream validation.

### 2.7. Protein Clustering to Find Regulatory RNA Binding

This work used the STRING database [16] (version 12.0) (https://string-db.org/ Data: 5 August 2024) to conduct experimental, text mining, neighborhood, gene fusion, co-expression, database, and co-occurrence methodologies for the prediction of protein–protein interaction (PPIs). A confidence score threshold of 0.4 was deemed medium in the research, which used Mus musculus as its reference species. Using Cytoscape (version 3.10.3), the Centiscape (version 1.3) plugin, and the Gephi software (version 0.9.2), we constructed the integrated network and analyzed it consistent with our prior work.

### 2.8. Enrichment Analysis

By using the enrichment analysis tool in the STRING app, we were able to better understand the network’s DEGs and the functional modules associated with each protein. Data from many sources, such as KEGG, WikiPathways, Reactome, and Gene Ontology (GO) keywords, were used to discover significant enrichment results in the study.

### 2.9. Identification of Key Co-Expression Modules Using WGCNA

The co-expression networks provide network-based gene screening methods for the purpose of discovering possible pathways for hSSC aging. We used the WGCNA R [17] program to construct gene co-expression networks using hSSC gene expression data profiles. Using the WGCNA approach, gene modules that show substantial correlations across samples were examined to see whether there was a link between these modules and the extrinsic characteristics observed in the samples. The pickSoftThreshold function was used to choose the soft powers β = 3 and 20 for the purpose of constructing a scale-free network. Next, the adjacency matrix was generated by entering the given values into the formula: aij = |Sij|H. The adjacency matrix between genes i and j is represented by aij, the similarity matrix produced from the Pearson correlation of all gene pairings is denoted by Sij, and the soft power value is shown by Σ. Afterwards, a topological overlap matrix (TOM) was generated using the adjacency matrix and its associated dissimilarity (1-TOM) [18]. Following that, hierarchical clustering was accomplished by means of a 1-TOM dendrogram. Using this dendrogram, we were able to associate genes with similar expression patterns and form gene co-expression modules. In order to identify functional modules in a co-expression network, we used the technique presented in a previous study to evaluate the links between modules and clinical features. Accordingly, modules that scored well in the correlation analysis were selected for further research since, according to clinical criteria, they were the most promising candidates. Our prior study provided a more comprehensive explanation of the WGCNA technique [4,9,19,20,21,22].

### 2.10. Machine Learning Algorithms

For differential gene expression (DGE) analysis, raw CEL files from GEO datasets were first normalized using the Robust Multi-array Average (RMA) method. To correct for technical variation across platforms, the ComBat function was applied for batch effect correction. An expression matrix was then constructed, and gene-level statistical testing was performed using empirical Bayes ANOVA. Genes with an absolute log2 fold change greater than or equal to 2 and an adjusted *p*-value below 0.05 were considered differentially expressed and selected for further analyses.

PPI analysis was performed by uploading the differentially expressed genes into the STRING database to construct an interaction network. The resulting network was imported into Cytoscape, where centrality measures such as degree, betweenness, and eigenvector were calculated to identify highly connected hub genes. Clustering of the network was then carried out using Gephi’s modularity algorithm, which allowed the identification of distinct functional modules representing coordinated protein groups.

For WGCNA, Pearson correlations were computed between all gene pairs to generate a similarity matrix. A soft-thresholding power (β = 12) was selected to approximate a scale-free network. The similarity matrix was transformed into a topological overlap matrix, and hierarchical clustering was applied to group genes into distinct co-expression modules. Each module was summarized by its eigengene, and correlations were calculated between module eigengenes and cellular phenotypes (SSC, ES-like, ESC). Modules showing strong correlations with pluripotency traits (r > 0.7, *p* < 0.01) were prioritized for hub gene discovery.

Finally, hub genes identified through WGCNA and PPI analyses were validated using single-cell RNA sequencing datasets (GSE149512). Expression profiles were normalized, and clusters of undifferentiated spermatogonia (THY1^+^ and ID4^+^) were identified. The candidate hub genes were then examined across these clusters, and genes consistently expressed in spermatogonial populations were confirmed as biologically relevant regulatory factors in SSC reprogramming.

### 2.11. Validation of the Hub Genes with scRNA-Seq Datasets in Regulatory RNA Binding

To validate the co-expression modules identified in Section 2.9, we analyzed hub genes using single-cell RNA sequencing (scRNA-seq) datasets. The WGCNA-based modules previously constructed from hSSC expression data were cross-referenced with publicly available scRNA-seq profiles. This approach allowed us to determine whether the candidate hub genes identified through co-expression analysis were consistently expressed in human spermatogonial populations. Modules that demonstrated strong correlations with clinical and biological features in Section 2.9 were prioritized for validation, ensuring that the selected genes had both computational and experimental support.

### 2.12. Validation of the Hub Genes with scRNA-Seq Datasets

For transcriptome analysis, the previous ten genomic datasets (GSE149512 [23]) were used. Learn everything about the methodology behind the creation of these datasets from the source articles and cultural dataset. In conclusion, both datasets used human spermatogonia. The “culture” scRNA-seq dataset may have originated from spermatogonia. Except for serving as research markers, the transgenes have little effect on the biology of the spermatogonia population. A dataset consisting of adult testis was assembled using fluorescence-activated cell sorting (FACS) and two distinct sets of markers: CD9Bright and ID4-eGFP+. Research on stem cells via transplantation has shown their reality. Undifferentiated spermatogonia cultures were first cultivated using the THY1+ subset of spermatogonia extracted from adult testes. The next step was to keep the cultures in an environment with optimal circumstances for glycolysis (10% O_2_, 5% CO_2_) for a period of 10 weeks, or 10 passes. Following the collection of spermatogonia from feeder cells, three separate rounds of single-cell RNA sequencing (scRNAseq) were carried out. The fact that stem cells can be cultured was shown by spermatogonial implantation. We have shown in the past that ID4-GFP+ spermatogonia may be successfully separated from adult mouse testis via THY1+ selection. Previous research has employed a variety of approaches to enhance spermatogonia, but the populations examined have been very consistent.

### 2.13. Analysis of Gene Expression Using Fluidigm qPCR

Fluidigm dynamic array chips were used for gene expression analysis in order to quantify SSCs, ES-like cells, and MEFs (Mouse Embryonic Fibroblasts, a control group). Exactly like our previous experiments. Here are the processes that were done utilizing the BioMark system: normalizing genes, selecting cells, transcribed messenger RNA (mRNA), amplified before using TaqMan qPCR for quantification. Two technical and three biological replicates made up each sample’s total of six. Data analysis was carried out using SPSS, Excel, and GenEx (v.7.0). GAPDH was used for standardization.

### 2.14. Statistical Analysis

To be sure the results could be reproduced, the trials were carried out three times. We used SPSS 27.0, the Statistical Package for the Social Sciences, to conduct our statistical analysis. The Shapiro–Wilk normalcy test was used to determine whether the gene expression data was normally distributed. A non-parametric test was used since the data was not normally distributed. To compare numerous independent groups, the Kruskal–Wallis H test was used. After that, we used the Bonferroni adjustment to account for multiple comparisons in our post hoc pairwise comparisons. Rather than using the Kruskal–Wallis test, normally distributed data were subjected to one-way ANOVA, with subsequent pairwise comparisons corrected using the Bonferroni method. A *p*-value less than 0.05 was used to establish statistical significance.

## 3. Results:

### 3.1. Isolation of SSCs and Derivation of ES-like Cells from SSCs

Using the Sox2 and Klf4, we were able to extract SSCs from adult mice testes. A single-cell solution was produced by enzymatically separating the seminiferous tubules using collagenase/dispase. These tubules were the first sites of the Oct4-GFP signal (Figure 1(A1–A3)). The next step was to grow the cells in mGSC medium. Emergence of SSCs occurred between days 2 and 14. SSCs are small, round, compact cells that cluster together to form colonies. Although early Klf4-GFP expression was low in several SSCs, this altered as the culture progressed. The Oct4-GFP signal, however, was reactivated after pluripotency induction. Colonies resembling embryonic stem cells (ES)—specifically, mESCs—became apparent throughout the course of the extended incubation. Characteristics of epiblast-derived cells included a high level of Oct4-GFP expression, a smooth edge profile, and spindle-to-round shape (Figure 1B).

### 3.2. Immunostaining Reveals Presence of Stemness Markers in ES-like Cells Compared to SSCs

We can see the expression of some genes in a qualitative way using the immunostaining approach. Immunohistochemistry (IHC) and immunocytochemistry (ICC) were both used in this investigation. Figure 2 shows the results of our ICC comparison of the OCT4 and DAZL gene expression rates in ES-like cells and SSCs. The stemness features of SSC-derived ES-like cells were shown by their positive OCT4 and SOX2 results in the ICC test. But ES-like cells do not express Nanog, SSEA4, or Ddx4, so they are probably not SSCs after all, and that is why they are distinct. In addition, the SSCs that tested positive for DAZL did not show any staining for VIM, which might indicate a link that has to be explored further. These results agree with our microarray study that showed different gene expression patterns for these proteins in the two types of cells.

### 3.3. Chimera Formation Analysis

Injecting GSCs into mouse blastocysts allowed us to study chimera generation and determine if ES-like cells are comparable to ESCs in their capacity to contribute to embryonic development in vivo. Using a micromanipulator, 10–15 single ES-like cells were inserted into 3.5-day-old mouse blastocysts. The next step was to inject the embryos into the uteruses of women who were artificially pregnant for the experiment. We harvested the embryos after they had gestated for 18 days. One way to tell whether a pup is chimerism is by looking at its coat color; some puppies had spots of both the host embryo’s and the ES-like cell clone’s coat color.

### 3.4. Pluripotency Markers in ES-like Cells Compared to SSCs

We utilized Transcriptome Analysis Console (version 4.0) (TAC) to identify genes that were differentially expressed between the ES-like cells group and the SSCs group. Using the RMA approach for normalization, we used gene expression criteria with a significance level of less than 0.05 and a Log2 fold change cut-off of less than or equal to −2. This criterion led to the identification of 3956 genes that showed differential expression (DEGs) between SSCs and ES-like cells (Figure 3A). Out of 3956 genes that showed differential expression, 1615 were first chosen based on specifications such a smaller *p*-value and a bigger Log2 fold change (−2 > Log2 fold change > 2). Out of 2145 DEGs, we applied a primary cutoff of log2 fold change ≥ |2| for stringent DEG selection. In addition, to capture a broader set of biologically relevant genes, a secondary analysis was performed at log2 fold change ≥ |2|, which identified additional candidates for downstream validation. The results showed that 1031 of the selected genes were upregulated and 586 were downregulated in ES-like cells. Notable DEGs include: Pou5f1, Ddx4, Ctdsp1, Gtf2f1, Ptbp1, Ctdspl, Ctdsp2, Ctdp1, Timm23, Timm21, Timm17b, Cdca3, Rest, Dusp11, Nif3l1, Zfp664, Mbp, Zp3, Sohlh2, Dppa3, Stra8, Sohlh1, Figla, Nobox, Dppa4, Utf1, Mael, Piwil2, and Nanos2. The fact that pluripotency-related genes like Nanog are being expressed at higher levels suggests that ES-like cells may exhibit stemness properties to a greater extent than SSCs. The downregulation of genes in ES-like cells (shown in Figure 3) suggests that these cells are repressing differentiation-related or lineage-specific pathways.

To make it easier to see the common DEGs across these studies, we developed a heatmap plot and a Venn diagram. The gene expression patterns of ESCs and ES-like cells are quite comparable, according to our findings. We aimed to compare a pluripotent cell type (ES-like cells) with SSCs, and while the similarities were not exact, they were enough to persuade us that the cells were pluripotent. While similarity in gene expression does offer a biological basis for DEG identification, it is insufficient on its own to guarantee statistical accuracy. By showing that the ES-like cells were in a pluripotent state, we confirmed that the DEGs recorded are actual differences between pluripotent cells and SSCs, and not just artifacts from poorly defined cells. The variable expression of key regulatory genes such as Tdgf1, Nanog, and Tet1 exemplifies the dynamic control of pathways critical for the maintenance of stem cell characteristics and the determination of its fate. This strong agreement backs up our identified DEGs, proves that ES-like cells maintain key pluripotent characteristics, and sets the stage for further studies along these lines. Curiously, when comparing ESCs with SSCs, all the genes that were found to have significantly different expression levels between the two cell types were also present: SALL4, SOHLH1, SOHLH2, STRA8, SYCP1, SYCP3, TCL1B, TDGF1, TDRD1, TDRD12, TDRD5, and TDRD6. Using the same selection and normalization criteria, we repeated the comparison study between ESCs and SSCs to confirm our results. Out of 2145 differentially expressed genes (DEGs), 2.3 were chosen for further analysis using strict *p*-value and fold-change criteria (−2 > Log2 fold change > 2).

### 3.5. Network Analysis Reveals 63 Hubs and Four Distinct Functional Protein Clusters from DEGs in Regulatory RNA Binding Genes

In order to construct a PPI network, we submitted the 1236 filtered DEGs across the test groups from the prior stage to the STRING database. The next step was to examine the built network using Cytoscape (v.3.6.0) in accordance with the specifications of the network. One of the most important concepts in network analysis is centrality, which helps to pinpoint which nodes are most important. There are several approaches to estimate the relevance of nodes. Degree, proximity, betweenness, and eigenvector centrality are some of the centrality measurements that provide information on a node’s function in a network. Consequently, the network’s 632 nodes were sifted for the most significant DEGs using several centrality metrics. To avoid reducing the breadth and depth of our route analysis study due to an inadequate number of nodes, we used node filtering to keep just the most crucial and applicable nodes in our dataset. Using this method, we can conduct an in-depth analysis while keeping the emphasis on the most important aspects. Using this strategy, we can pinpoint the genes that are absolutely necessary for the connection of networks. Four separate protein clusters (Modules) were generated from a network of seventy hub proteins using Gephi’s in-built modularity algorithm. There are a total of 21 nodes in Cluster 1, 16 in Cluster 2, 12 in Cluster 3, and 21 in the final cluster (module). In the process of SSC differentiation into ES-like cells, each cluster, or module, represents proteins that cooperate to carry out certain tasks. Because of this, we can investigate enriched pathways involving all of the network’s DEGs and learn more about the roles played by specific proteins and the genes that interact closely with them along these pathways (Figure 4).

### 3.6. WGCNA Identifies Key Co-Expression Modules Associated with SSC Reprogramming

To explore gene co-expression patterns during SSC reprogramming, we constructed weighted gene co-expression networks using the WGCNA algorithm. First, we determined the appropriate soft-thresholding power (β) to achieve a scale-free topology. As shown in Figure 5, β = 12 provided the best balance between scale independence (R^2^ > 0.85) and mean connectivity and was therefore selected for network construction. Using this threshold, hierarchical clustering of genes based on topological overlap was performed. The resulting dendrogram revealed multiple distinct co-expression modules, each represented by a different color. Next, we correlated these modules with biological traits of interest (SSC, ES-like cells, and ESCs). The module–trait relationship heatmap demonstrated that the blue and brown modules were most strongly correlated with pluripotency traits (correlation > 0.7, *p* < 0.01). These modules were therefore prioritized for downstream hub gene identification and validation. Finally, representative eigengene expression patterns confirmed that genes within the selected modules exhibited coherent expression across cell types. The identification of these modules provided the foundation for subsequent hub gene analysis using scRNA-seq datasets (Section 2.10).

### 3.7. Machine Learning

Differential gene expression analysis identified a total of 3956 genes that were significantly altered between SSCs and ES-like cells. Of these, 1031 genes were upregulated and 586 were downregulated in ES-like cells. Key pluripotency-associated genes, including Pou5f1, Nanog, Dppa3, Utf1, and Stra8, were among the upregulated genes, confirming that ES-like cells had acquired stemness features at the transcriptional level.

Network-based approaches further refined these results. Protein–protein interaction analysis of the differentially expressed genes yielded a network consisting of 632 nodes. Centrality-based ranking identified 63 hub genes, which were distributed across four functional protein clusters. These clusters were enriched for biological processes such as RNA helicase activity, piRNA metabolic processing, DNA methylation, and regulatory RNA binding. Notably, Ctdsp1, Rest, and Stra8 emerged as central RNA-binding regulators potentially driving SSC reprogramming.

WGCNA provided additional insights into coordinated gene regulation. Using a soft threshold of β = 12, hierarchical clustering revealed multiple gene co-expression modules. Correlation analysis between module eigengenes and cell type traits showed that the blue and brown modules were most strongly associated with pluripotency (correlation > 0.7, *p* < 0.01). These modules contained many of the same regulatory genes identified in the DEG and network analyses, highlighting their robustness as candidates for further validation.

Finally, validation using publicly available single-cell RNA sequencing datasets confirmed the expression of candidate hub genes across human spermatogonial populations. Specifically, Ctdsp1, Rest, and Stra8 were consistently expressed in undifferentiated THY1^+^ and ID4^+^ spermatogonia, demonstrating conservation of regulatory mechanisms between mouse and human. This cross-validation reinforced the biological significance of the computationally predicted hub genes and confirmed their potential role in SSC reprogramming (Figure 5).

### 3.8. Enrichment Analysis in ES-like Cell Function

In order to get accurate and thorough findings, we ran separate enrichment studies on each protein cluster (module). Using the STRING database (Mus musculus) and Gene Ontology (GO), KEGG, Reactome, and Wikipathways, we built protein–protein interaction (PPI) networks and conducted functional enrichment studies on the genes within each cluster. We gave the top 10 pathways more importance according to their strength and False Discovery Rate (FDR) (*p* < 0.05). To get more biological insights, we looked at pathways with lower Adjusted *p*-values. Our method allowed us to zero in on the most essential paths while also considering less obvious but potentially consequential results (Figure 6). Go:0034587 piRNA metabolic process, GO:0006305 DNA alkylation, and GO:0006306 DNA methylation are abundant in the second cluster, which is important in gamete production. This group is highly linked to GO:0003724 RNA helicase activity, GO:0061980 regulatory RNA binding, GO:0034584 piRNA binding, GO:1905538 polysome binding, and GO:0008186 ATP-dependent activity acting on RNA. This cluster probably regulates cellular plasticity and niche remodeling, which might have consequences in stem cell niches (Figure 6), considering that the relationship between collagen production, degradation, and integrin-mediated interactions are analyzed by GO analysis.

### 3.9. Testicular Cell Composition During Testis Development in Humans

At1,2, and 7 years of age, we used single-cell RNA sequencing in conjunction with publicly available information to describe the cellular diversity of the testis in humans. We used a testis maturation developmental timeline that we made up to compare different age groups. Many different age groups and developmental phases were represented in the datasets, beginning with prenatal (embryonic weeks 6–16), continuing through neonatal (postnatal days 2–7), prepubertal (11, 13, 14, 17, and 25 years old), and finally peri- and postpubertal (ages 13–25). The total number of testicular cells retained for future study was 82,220. Using well-established markers for testicular cells, UMAP embedding of the combined datasets found 17 different clusters of cells (Figure 1). Germ cells (DDX4+, ID4+), Sertoli cells (AMH+, SOX9+), myoid cells (ACTA2+, RGS5+), macrophages (CD14+), and endothelial cells (PECAM1+) are shown in Figure 7. Leydig cells and somatic precursors were detected in clusters 13, 14, and 15 due to the co-expression of DLK1 and NR2F2, as shown in Figure 6. The composition of germ cells changed with age, however throughout embryonic (W6–W16), neonatal (2D–7D), and prepubertal (1Y–7Ys) phases, the somatic cell makeup (Sertoli, Leydig, and somatic precursor) was more predominant than the germ cell composition. Figure 6 shows that throughout puberty, the number of germ cell stages, namely clusters 2–4, increased. As the testis develops, the cell composition changes to reflect the anatomical and functional changes in germline and somatic cells. Figure 8 shows that Nanog, Ddx4, Sall4, and Dppa5 gene expression changes when SSCs are converted to ES-like.

### 3.10. Real-Time PCR

There was quantitative and qualitative confirmation of our microarray gene expression findings via in vitro tests. We can find out whether our enriched pathways are robust and if our observed DEGs are valid using this strategy. To compare the mRNA expression rates of SSCs and ES-like cells, a quantitative method known as Fluidigm qPCR was used. Based on our Fluidigm gene expression data, we observed that ES-like cells and SSCs expressed varying quantities of Ddx4, Ctdsp1, Rest, Stra8, Dazl, and Klf4 (Figure 9). Inconsistencies in gene expression lend credence to our computational study’s findings. The Vim expression patterns, on the other hand, were not substantially different across the two types of cells.

### 3.11. Human Testicles Develop and Cell Composition

We integrated the results of single-cell RNA sequencing with data from publically accessible sources to define the cellular diversity of the testis throughout human development. The subjects were boys aged 1, 2, and 7 years. We compared various age groups using a developmental timetable we created for testis maturation. The datasets included a wide range of ages and stages of development, from prenatal (embryonic weeks 6–16) to neonatal (postnatal days 2–7) to prepubertal (11, 13, 14, 17, and 25 years) to peri- to postpubertal (ages 13–19). In all, 82,220 distinct testicular cells were set aside for further analysis (Figure 10).

## 4. Discussion

This study aimed to explore the transcriptional and functional transition of SSCs into ES-like cells, leveraging both in vitro culture systems and integrated transcriptomic analyses. Our results provide compelling evidence that SSCs derived from OCT4-GFP transgenic mice can be successfully reprogrammed into ES-like cells expressing core pluripotency markers and capable of contributing to early embryonic development. Through a combination of microarray profiling, protein–protein interaction analysis, WGCNA-based network modeling, and scRNA-seq validation, we have identified key regulatory genes and pathways that may govern this unique reprogramming process.

One of the core findings in our study was the upregulation of pluripotency-associated genes such as OCT4, NANOG, and SOX2 in SSC-derived ES-like cells, aligning with previous reports that SSCs can acquire pluripotent characteristics under defined conditions without exogenous gene transduction [24]. Our use of an OCT4-GFP reporter model further confirmed the activation of endogenous pluripotency programs. Notably, this transition occurred within 41–125 days of culture, indicating a gradual and dynamic reprogramming process, likely influenced by microenvironmental cues and epigenetic remodeling.

Our transcriptomic comparisons, using publicly available datasets (GSE43850 and GSE38776), revealed hundreds of DEGs between SSCs, ES-like cells, and ESCs. Among these, we observed significant enrichment in pathways related to cell cycle regulation, Wnt signaling, and chromatin remodeling—all of which have been previously implicated in pluripotency acquisition and maintenance [25,26]. For example, Wnt/β-catenin signaling is known to support self-renewal in both SSCs and ESCs, and its activation may serve as a molecular bridge during the SSC-to-ES-like transition [27,28].

PPI network analysis highlighted several hub genes, including SALL4, DPPA4, and UTF1, as central regulators. These genes not only function in early embryonic development but are also known to participate in reprogramming and chromatin accessibility. Such hub genes represent promising biomarkers or targets for optimizing SSC-derived pluripotency protocols [29]. This is consistent with recent network-based studies identifying SALL4 as a pivotal factor linking germline and pluripotent cell fates [30,31,32].

The application of Weighted Gene Co-expression Network Analysis (WGCNA) allowed us to identify modules of co-expressed genes significantly associated with the acquisition of pluripotency. Notably, the modules enriched in our ES-like cells overlapped with gene signatures involved in stem cell identity, metabolic reprogramming, and transcriptional regulation. These findings support the concept that SSC reprogramming is not driven by isolated genes but by coordinated regulatory modules, a notion reinforced by Jiang et al., who reported similar modular behaviors in organoid-derived stem cells [15,33].

To further validate our findings, we analyzed scRNA-seq datasets (GSE149512) representing human spermatogonial populations cultured under glycolytic conditions. Our identified hub genes were expressed in undifferentiated THY1^+^/ID4^+^ spermatogonia, supporting their relevance across species. These results not only validate the conservation of reprogramming-associated genes but also emphasize the translational potential of SSC research in regenerative medicine and fertility preservation. The Fluidigm-based qPCR analysis confirmed differential gene expression between SSCs, ES-like cells, and control MEFs, reinforcing the transcriptomic shifts observed in our microarray and network analyses. Together, these findings underscore the reproducibility and robustness of our experimental system.

Despite these advances, our study has several limitations. First, while our ES-like cells exhibited hallmarks of pluripotency and teratoma formation, their contribution to germline transmission in vivo remains to be tested. Second, although WGCNA and PPI analyses provide strong statistical associations, functional experiments such as gene knockdown or overexpression are necessary to confirm causality. Future studies should explore epigenetic landscapes (e.g., DNA methylation and histone modifications) to complement our transcriptomic insights.

In conclusion, our findings deepen the understanding of SSC reprogramming into pluripotent-like states by identifying key genes, pathways, and regulatory modules involved in this transformation. These insights not only advance basic stem cell biology but also hold potential for therapeutic strategies in infertility treatment, in vitro gametogenesis, and disease modeling using patient-derived SSCs.

## 5. Conclusions

This study provides new insights into the reprogramming potential of spermatogonial stem cells (SSCs) and their ability to acquire pluripotent characteristics under defined culture conditions. By utilizing OCT4-GFP reporter mice, transcriptomic profiling, protein interaction networks, and single-cell RNA-seq validation, we demonstrated that SSCs can be transformed into embryonic stem-like (ES-like) cells that express key pluripotency markers and contribute to early embryonic development. Our integrative approach identified core regulatory genes and co-expression modules that orchestrate this reprogramming process. The discovery of overlapping molecular signatures between SSCs and ESCs highlights the intrinsic plasticity of SSCs and supports their potential as an alternative, ethically less contentious source of pluripotent cells. These findings not only enhance our understanding of germline stem cell biology but also open promising avenues for clinical applications in regenerative medicine, reproductive biology, and infertility treatment. Future research should focus on validating the functional roles of identified hub genes and evaluating the germline competency of SSC-derived ES-like cells in vivo.

## Figures and Tables

**Figure 1 cells-14-01632-f001:**
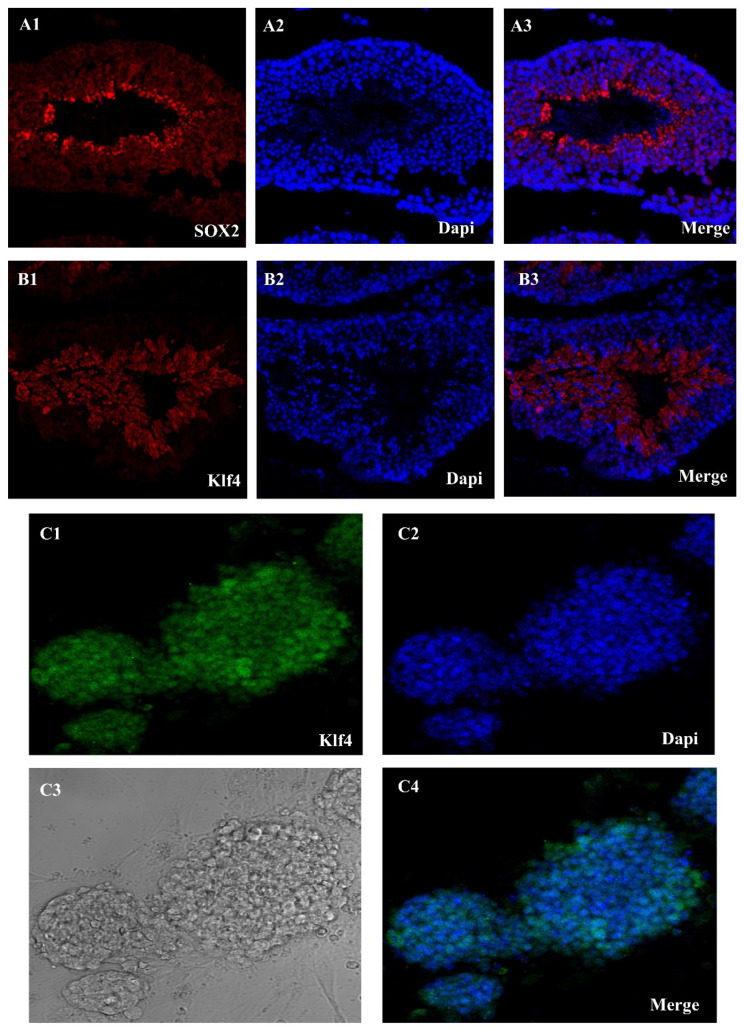
Sox2-Klf4-GFP expression in seminiferous tubules and Klf4 expression in SSC in cell culture. (**A1**–**A3**) Seminiferous tubules of an Oct4-GFP reporter adult transgenic mouse show Sox2-GFP expression, (**B1**–**B3**) Klf4-GFP expression in seminiferous tubules, and (**C1**–**C4**) SSC culture oct4-GFP reporter adult transgenic mouse show Klf4-GFP expression in a mouse ESC culture medium (Scale bar = 50 μm).

**Figure 2 cells-14-01632-f002:**
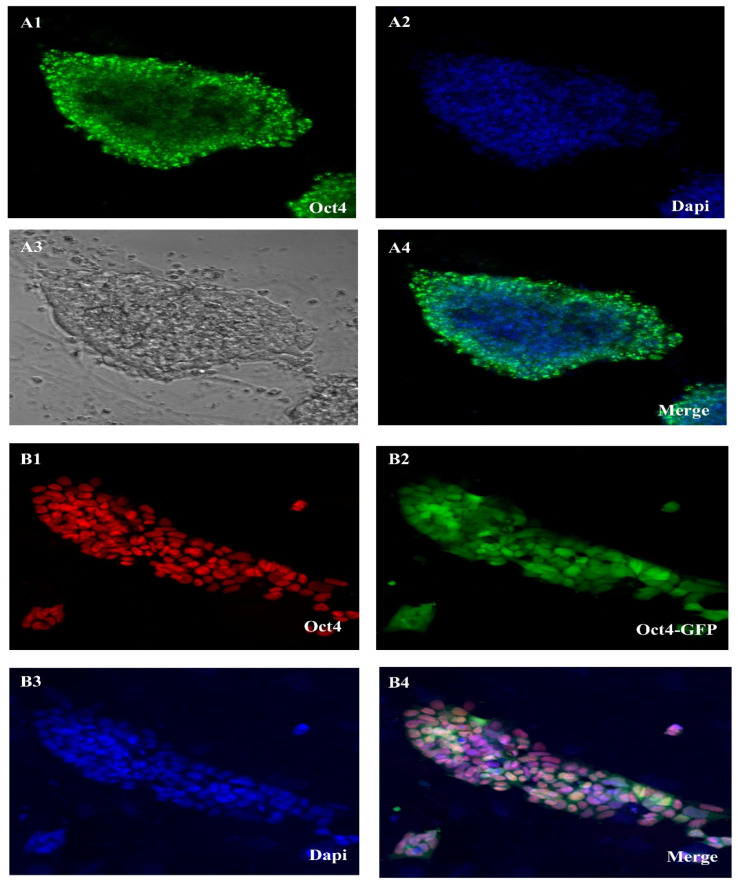
Oct4 expression in SSC in cell culture and Oct4 gene expression in ES-like. (**A1**–**A4**) SSC in cell culture of an Oct4-GFP reporter adult transgenic mouse shows Oct4-GFP expression and (**B1**–**B4**) Es-like of an Oct4-GFP reporter adult transgenic mouse shows Oct4-GFP expression (Scale bar = 50 μm).

**Figure 3 cells-14-01632-f003:**
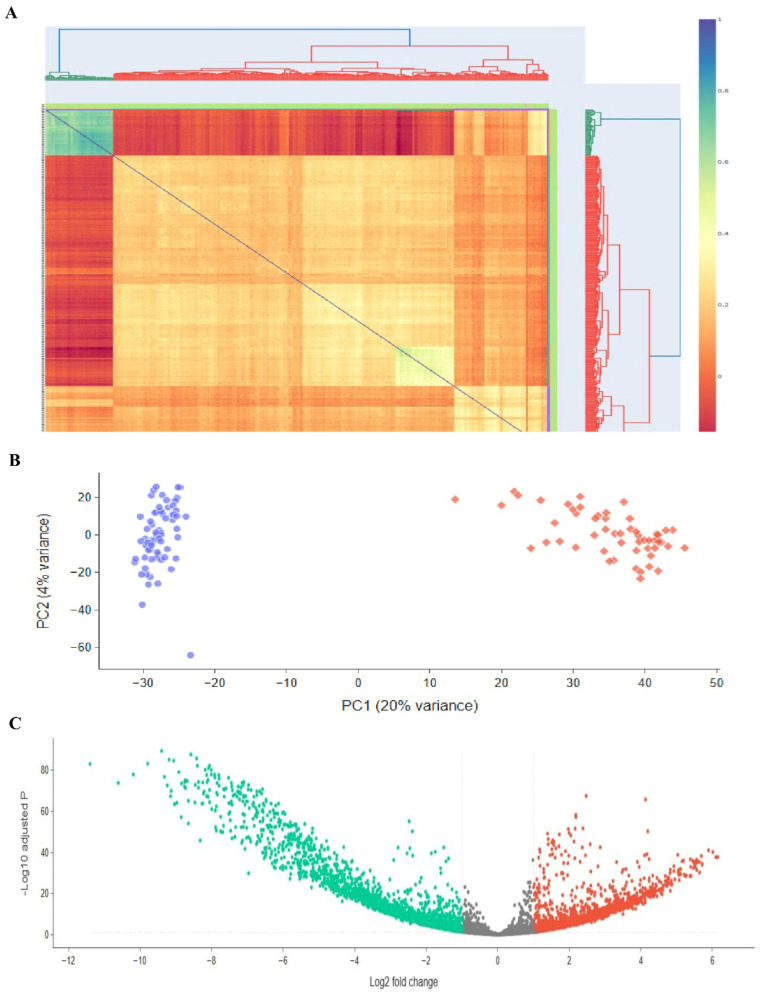
Investigating transcriptomics of embryonic stem cells, stromal cells, and ES-like cells. Publicly available microarray datasets GSE38776, which included SSCs, and GSE43850, which had ESCs and ES-like cells, were used to perform the differential gene expression research depicted in this figure. (**A**) A heatmap with clustering patterns showing all samples, including 3 SSCs from GSE38776 and 3 ESCs and 3 ES-like cells from GSE43850. There are some parallels in transcription between ES-like cells, SSCs, and ESCs, but there are also significant differences. (**B**) Genes with statistically significant changes in gene expression (*p*-value < 0.05) are highlighted in a volcano plot that shows differentially expressed genes (DEGs) between embryonic stem cells (ESCs) and stromal stem cells (SSCs). (**C**) The volcano map shows the Log2 fold change and *p*-values for DEGs between ES-like cells and SSCs, with a *p*-value threshold of less than 0.05. The results of the microarray investigation that examined the global expression patterns of SCs and ES-like cells are summarized below.

**Figure 4 cells-14-01632-f004:**
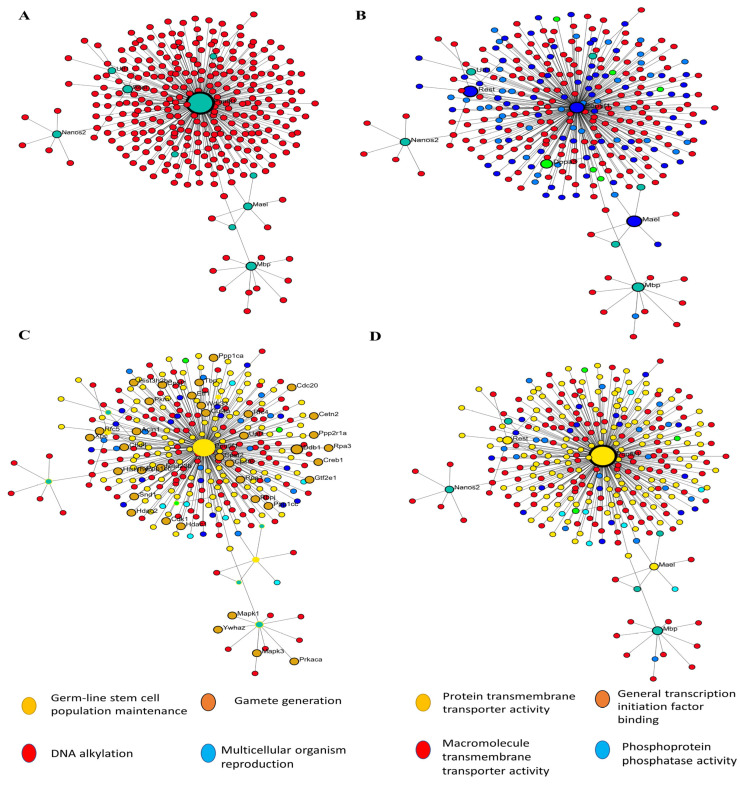
PPI and GO analysis. (**A**) PPI analysis transfer SSCs to ES-Like. (**B**) Finding hub gene involved, (**C**) biological process, and (**D**) Molecular function.

**Figure 5 cells-14-01632-f005:**
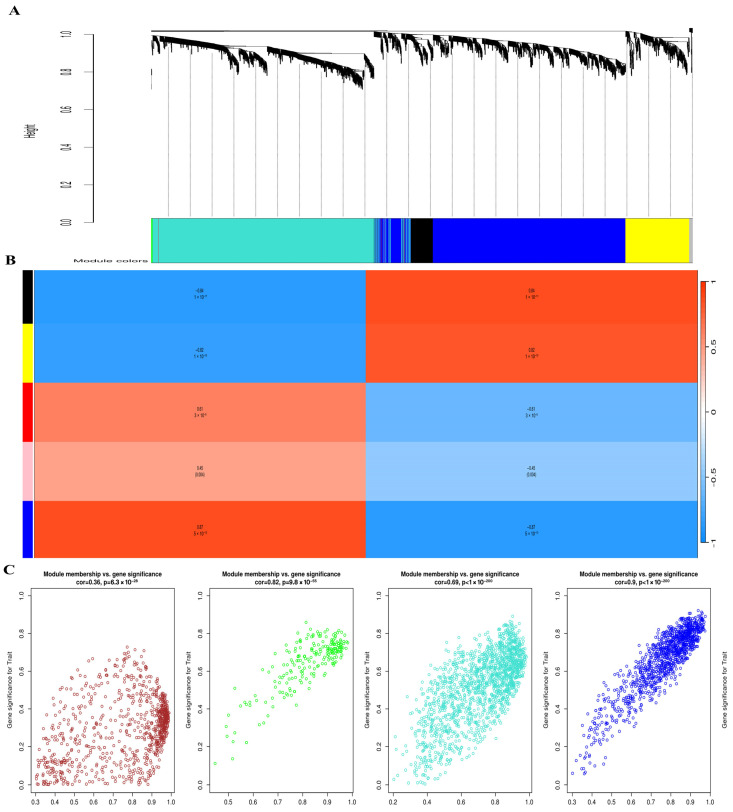
Weighted Gene Co-expression Network Analysis (WGCNA) of SSC reprogramming transcriptomes. (**A**) Hierarchical clustering dendrogram of genes showing distinct co-expression modules, each assigned a unique color. Modules represent groups of genes with highly correlated expression patterns. (**B**) Module–trait relationship heatmap displaying correlations between identified modules and cellular phenotypes (SSC, ES-like, and ESCs). Red indicates positive correlation, while blue indicates negative correlation; the strength of the correlation is shown by color intensity. Modules strongly associated with pluripotency traits are highlighted. (**C**) Scatter plots showing the relationship between module membership and gene significance within key modules. High correlations indicate that genes with strong module connectivity also contribute significantly to pluripotency-related traits, confirming the biological relevance of the identified modules.

**Figure 6 cells-14-01632-f006:**
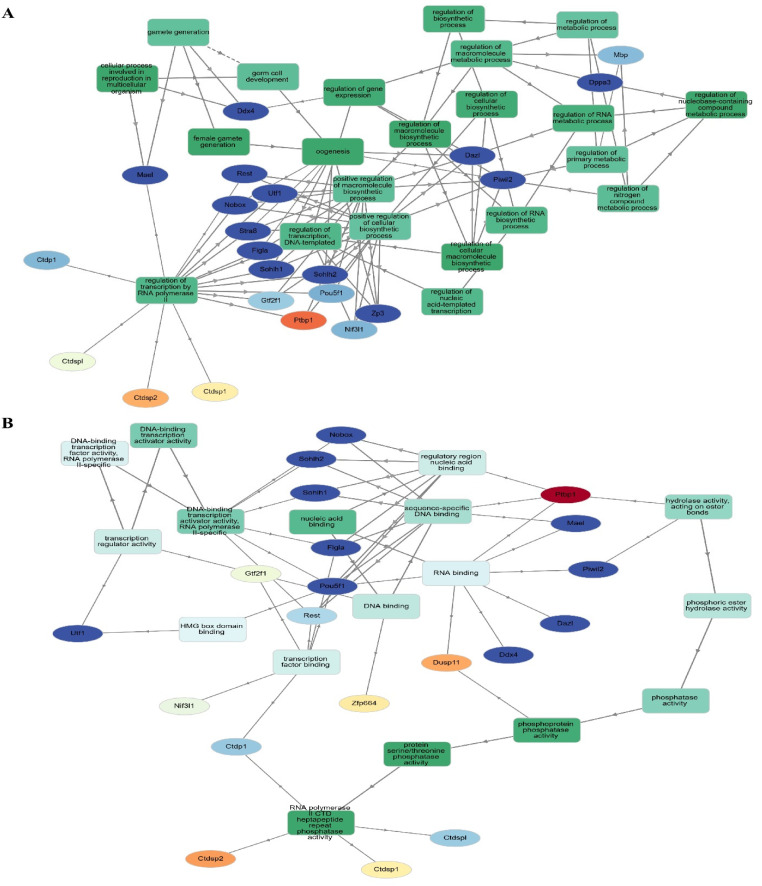
Four separate functional clusters are revealed by Go analysis. (**A**) The PPI network was constructed and assessed using Gephi. By using clustering algorithms and filters, we were able to identify nearby hub proteins and the relationships between them. Different kinds of functional groupings are shown by the color-coding of the network into four distinct clusters. Larger nodes, indicative of a higher degree of connectivity and maybe indicating the regulatory importance of each cluster, reflect hub proteins with many connections. (**B**) As shown in this picture, hub proteins have the potential to coordinate crucial biological functions along their routes.

**Figure 7 cells-14-01632-f007:**
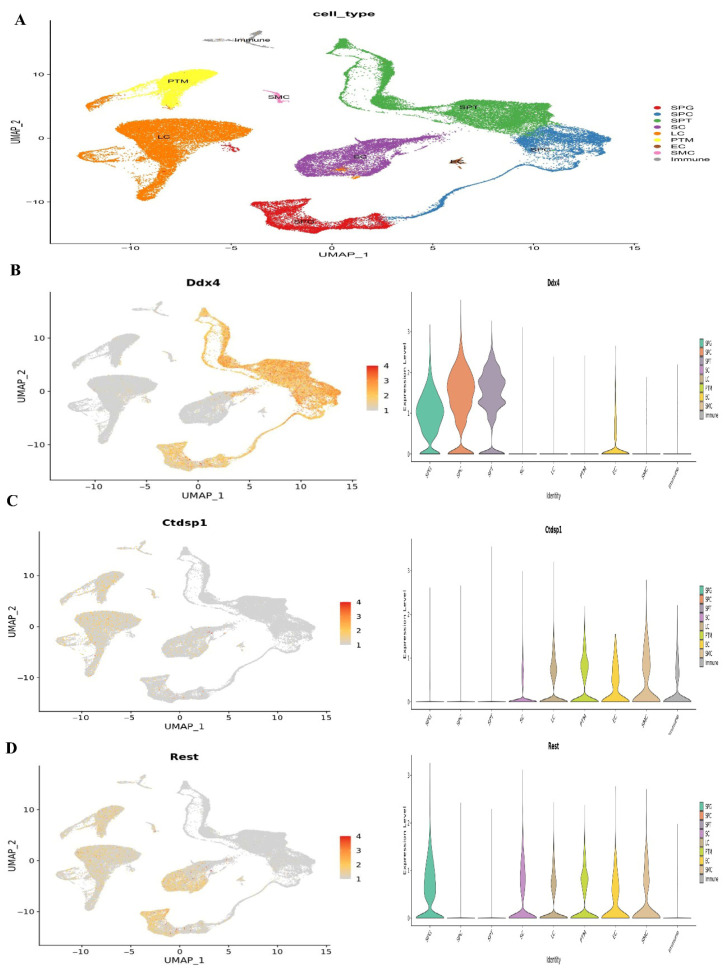
Single-cell RNA sequencing investigation of mouse adult spermatogonia. (**A**) UMAP plot depiction of germ cells from merged single-cell RNA sequencing data, (**B**) Ddx4 gene expression, (**C**) Ctdsp1 and (**D**) Rest gene expression.

**Figure 8 cells-14-01632-f008:**
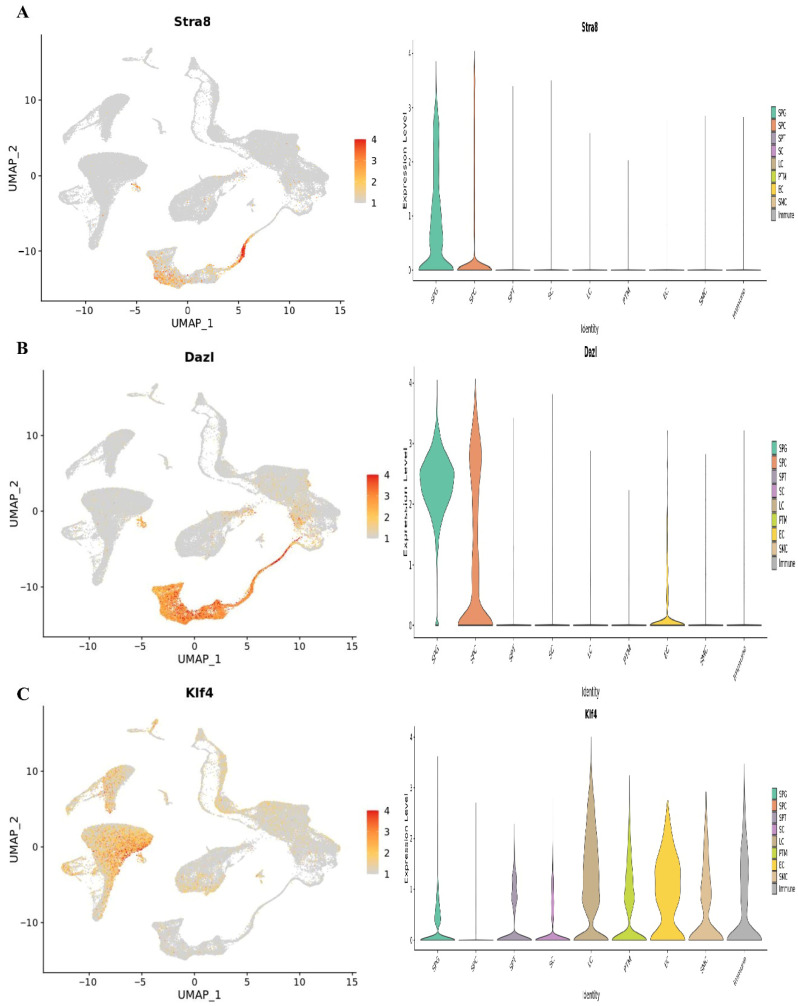
The UMAP plots classify testicular cells according to their age. Germ cell clusters in UMAP are also color-coded according to age, and the expression patterns of important markers for spermatogonia, spermatocytes, and spermatids are reflected in the plots. The expression of the (**A**) Stra8, (**B**) Dazl, and (**C**) Klf4.

**Figure 9 cells-14-01632-f009:**
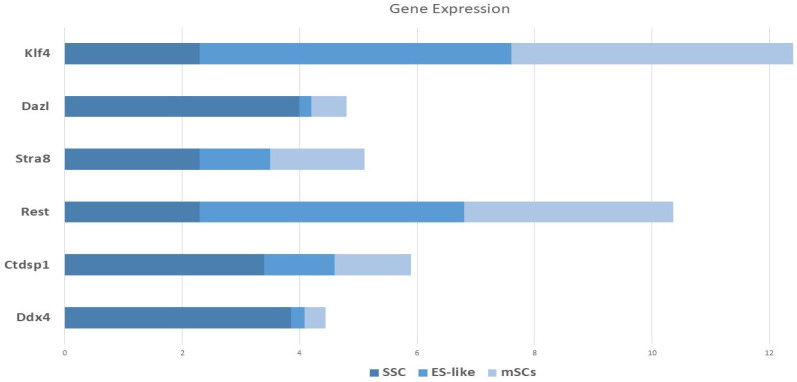
Monitoring the progression of stem cell lines into endothelial-like cells using gene expression profiling using Fluidigm qPCR. A panel of germline-associated and pluripotency genes’ relative Log2 fold change in mRNA expression during spermatogonial stem cell (SSC) reprogramming into ES-like cells is shown on the *Y*-axis of this image. Expression levels were normalized to those of MEFs, which are embryonic fibroblasts from mice.

**Figure 10 cells-14-01632-f010:**
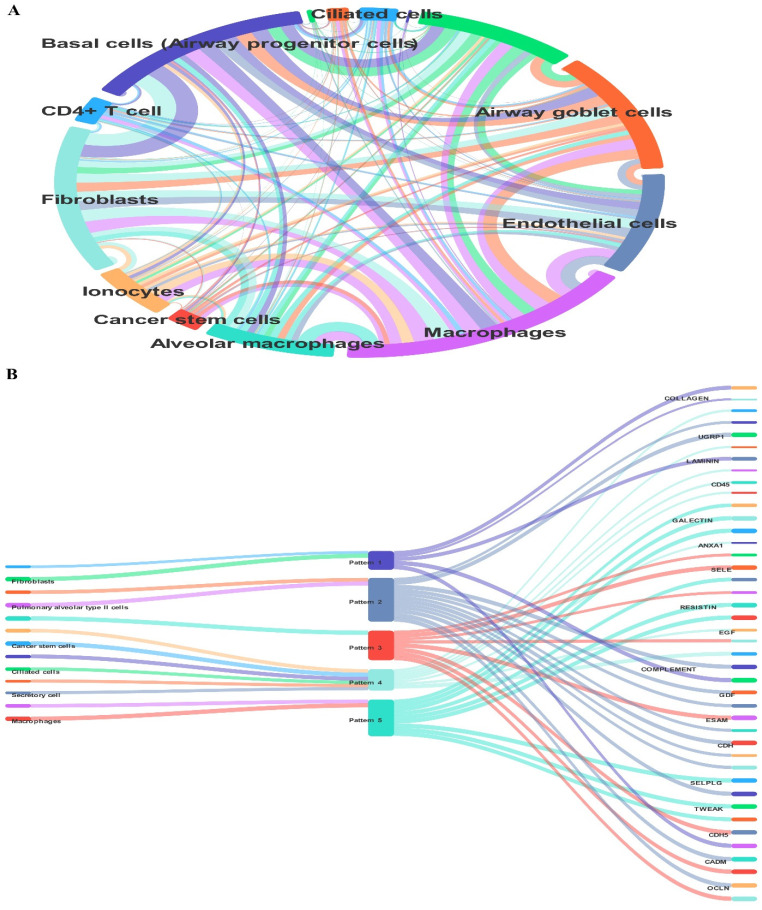
Thorough analysis of the germ cell lineage transcriptome and exploration of the human testis single-cell transcriptome. (**A**) Testicular cells from the prenatal to postnatal phases of testicular development are shown in the UMAP map, which is obtained from integrated single-cell RNA sequencing data. (**B**) The 82,220 testicular cells are organized into 17 clusters, each with its own color code, which reflect different germ cell and somatic cell lineages.

## Data Availability

The original contributions presented in this research are included in the article.

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
