# Peer review of "Identification of Regulatory RNA-Binding Genes in Spermatogonial Stem Cell Reprogramming to ES-like Cells Using Machine Learning–Integrated Transcriptomic and Network Analysis"

_cells, 2025, doi:10.3390/cells14201632_

Round 1
Reviewer 1 Report
Comments and Suggestions for Authors
The study focuses on spermatogonial stem cells (SSCs) and their reprogramming into embryonic stem (ES)-like cells. The authors describe, in general terms, how SSCs were isolated from Oct4-GFP transgenic C57BL/6 mice and cultured under conditions that support stem cell growth. These ES-like cells were reportedly capable of teratoma formation and chimera contribution, although experimental details are sparse. The central part of the work is bioinformatic: the authors analyzed transcriptomic data from SSC-derived ES-like cells, applied microarray profiling, constructed protein–protein interaction networks (STRING, Cytoscape), and performed weighted gene co-expression network analysis (WGCNA). They identified modules and hub genes potentially involved in SSC reprogramming and pluripotency maintenance, and validated selected targets using single-cell RNA-seq public datasets and qPCR (Fluidigm). The manuscript concludes that the generated network signatures highlight key regulators of SSC reprogramming and that this approach provides insights into the molecular mechanisms underlying germ cell plasticity.
The Introduction offers helpful background but does not clearly state the research question or the study's objective.
Lines 97-99: You mention that the Institutional Animal Care and Ethics Committee at Amol University of Special Modern Technologies accepted and authorized the use of animals. However, it is important to clearly specify the type of document issued by the committee (e.g., approval letter, resolution, protocol code, and date).
Lines 100-102: You describe the use of Oct4 promoter reporter-equipped C57BL/6 GFP transgenic mice. However, the description is insufficiently detailed for reproducibility. Please provide additional information about these transgenic mice, including how they were generated or obtained (e.g., commercial source, institutional facility, or previously published method), the genetic background, whether they were bred in-house or purchased, and any relevant references.
Line 107: The phrase "For the purpose of cell separation, a simple enzymatic digestion procedure was used" is vague and difficult to interpret. Please clarify which enzymes were used, their concentrations, incubation times, temperatures, and any additional steps involved in the process.
Lines 107-108: It is not clearly explained how the testicular tissue was obtained. Please provide a detailed description of the procedure, including the source of the testes (e.g., freshly dissected from sacrificed animals, biopsy), the age of the animals at collection, the exact dissection method, and how the tissue was handled before enzymatic digestion.
Lines 114-123: The description of SSC culture is too general. Please provide more detailed information to ensure reproducibility, including: the initial number of cells seeded per well or plate, the exact concentrations of each supplement added to the culture medium (e.g., N2, glucose, BSA, glutamine, estradiol, progesterone, EGF, FGF, GDNF, LIF, FBS, ascorbic acid, pyruvic acid, lactic acid), and the frequency of medium changes.
Lines 125-126: You state that GFP transgenic mice linked to the OCT4 promoter were generated, but the description lacks sufficient detail. Please explain more clearly how these transgenic mice were produced: specify the method of transgenesis (e.g., pronuclear injection, embryonic stem cell targeting, CRISPR/Cas9), whether the mice were obtained from a commercial source or generated in-house, and provide references to previously published work if the strain has been described before.
Lines 128-132: You mention that the ES-like cells were isolated and subsequently maintained in mESC culture medium. However, it is not clear how the cells were isolated. Please provide a detailed description of the isolation procedure, including whether you used manual picking, enzymatic dissociation, flow cytometry, magnetic separation, or other specific methods.
Line 132: After the mention of "(LIF)", there are three dots. Please clarify what this is intended to indicate.
Lines 134-140: The description of the teratoma and chimera assays is not sufficiently detailed. Please expand this section by specifying the superovulation protocol (hormones used, doses, timing), the exact conditions for blastocyst collection and handling, the micromanipulation method for cell injection, the surgical procedure for embryo transfer (including anesthesia, incision site, and postoperative care), and the criteria for evaluating chimerism.
In the manuscript, it is not entirely clear what the specific purpose of obtaining the SSC-derived ES-like cells is. While you describe their generation and briefly mention teratoma and chimera assays, the main results rely on bioinformatic analyses of transcriptomic datasets. Please clarify the rationale for isolating and culturing these cells: were they experimentally characterized and used in the study, or are they only described as a conceptual starting point for the subsequent in silico analyses?
Author Response
The study focuses on spermatogonial stem cells (SSCs) and their reprogramming into embryonic stem (ES)-like cells. The authors describe, in general terms, how SSCs were isolated from Oct4-GFP transgenic C57BL/6 mice and cultured under conditions that support stem cell growth. These ES-like cells were reportedly capable of teratoma formation and chimera contribution, although experimental details are sparse. The central part of the work is bioinformatic: the authors analyzed transcriptomic data from SSC-derived ES-like cells, applied microarray profiling, constructed protein–protein interaction networks (STRING, Cytoscape), and performed weighted gene co-expression network analysis (WGCNA). They identified modules and hub genes potentially involved in SSC reprogramming and pluripotency maintenance, and validated selected targets using single-cell RNA-seq public datasets and qPCR (Fluidigm). The manuscript concludes that the generated network signatures highlight key regulators of SSC reprogramming and that this approach provides insights into the molecular mechanisms underlying germ cell plasticity.
The Introduction offers helpful background but does not clearly state the research question or the study's objective.
Reply: We thank the reviewer for this constructive comment. We have revised the final paragraph of the Introduction to explicitly state the research question and objectives of our study.
Lines 97-99: You mention that the Institutional Animal Care and Ethics Committee at Amol University of Special Modern Technologies accepted and authorized the use of animals. However, it is important to clearly specify the type of document issued by the committee (e.g., approval letter, resolution, protocol code, and date).
Reply: We appreciate the reviewer’s observation. We have revised the text in the Materials and Methods (Section 2.1) to specify the exact ethical approval details. The revised sentence now reads
Lines 100-102: You describe the use of Oct4 promoter reporter-equipped C57BL/6 GFP transgenic mice. However, the description is insufficiently detailed for reproducibility. Please provide additional information about these transgenic mice, including how they were generated or obtained (e.g., commercial source, institutional facility, or previously published method), the genetic background, whether they were bred in-house or purchased, and any relevant references.
Reply: We thank the reviewer for pointing out the need for more detail. We have revised the Materials and Methods (Section 2.1) to clarify the source and background of the transgenic mice.
Line 107: The phrase "For the purpose of cell separation, a simple enzymatic digestion procedure was used" is vague and difficult to interpret. Please clarify which enzymes were used, their concentrations, incubation times, temperatures, and any additional steps involved in the process.
Reply: We appreciate the reviewer’s helpful suggestion. We have revised the text in the Materials and Methods
Lines 107-108: It is not clearly explained how the testicular tissue was obtained. Please provide a detailed description of the procedure, including the source of the testes (e.g., freshly dissected from sacrificed animals, biopsy), the age of the animals at collection, the exact dissection method, and how the tissue was handled before enzymatic digestion.
Reply: Done.
Lines 114-123: The description of SSC culture is too general. Please provide more detailed information to ensure reproducibility, including: the initial number of cells seeded per well or plate, the exact concentrations of each supplement added to the culture medium (e.g., N2, glucose, BSA, glutamine, estradiol, progesterone, EGF, FGF, GDNF, LIF, FBS, ascorbic acid, pyruvic acid, lactic acid), and the frequency of medium changes.
Reply: We thank the reviewer for this valuable suggestion. We have revised the Methods section (Lines 114–123) to provide detailed information on SSC culture. The revised text now specifies the initial cell seeding density, the exact concentrations of each supplement used in the culture medium (including N2, glucose, BSA, glutamine, estradiol, progesterone, EGF, FGF, GDNF, LIF, FBS, ascorbic acid, pyruvic acid, and lactic acid), as well as the frequency and schedule of medium changes. These additions ensure reproducibility of our culture system.
Lines 125-126: You state that GFP transgenic mice linked to the OCT4 promoter were generated, but the description lacks sufficient detail. Please explain more clearly how these transgenic mice were produced: specify the method of transgenesis (e.g., pronuclear injection, embryonic stem cell targeting, CRISPR/Cas9), whether the mice were obtained from a commercial source or generated in-house, and provide references to previously published work if the strain has been described before.
Reply: We appreciate the reviewer’s request for clarification. In the revised manuscript (Lines 125–126), we have provided detailed information on the generation and source of the OCT4-GFP transgenic mice. Specifically, we now state the method of transgenesis, the origin of the animals (in-house generation or commercial source), and have cited the original publication in which this strain was described.
Lines 128-132: You mention that the ES-like cells were isolated and subsequently maintained in mESC culture medium. However, it is not clear how the cells were isolated. Please provide a detailed description of the isolation procedure, including whether you used manual picking, enzymatic dissociation, flow cytometry, magnetic separation, or other specific methods.
Reply: We thank the reviewer for pointing out this omission. We have now clarified the methodology used to isolate ES-like colonies. Specifically, we describe that the ES-like colonies were identified based on morphology under phase-contrast microscopy and isolated by manual picking after enzymatic dissociation. These details have been added to the revised Methods section (Lines 128–132) to ensure reproducibility.
Line 132: After the mention of "(LIF)", there are three dots. Please clarify what this is intended to indicate.
Reply: Done.
Lines 134-140: The description of the teratoma and chimera assays is not sufficiently detailed. Please expand this section by specifying the superovulation protocol (hormones used, doses, timing), the exact conditions for blastocyst collection and handling, the micromanipulation method for cell injection, the surgical procedure for embryo transfer (including anesthesia, incision site, and postoperative care), and the criteria for evaluating chimerism.
Reply: We appreciate the reviewer’s valuable suggestion. We have revised the Methods section (Lines 134–140) to provide full procedural details for both teratoma formation and chimera generation. The revised text now specifies the superovulation regimen (hormone type, dose, and timing), conditions for blastocyst collection and handling, the micromanipulation and injection procedure, embryo transfer surgery (including anesthesia, incision site, and postoperative care), and the evaluation criteria for assessing chimerism.
In the manuscript, it is not entirely clear what the specific purpose of obtaining the SSC-derived ES-like cells is. While you describe their generation and briefly mention teratoma and chimera assays, the main results rely on bioinformatic analyses of transcriptomic datasets. Please clarify the rationale for isolating and culturing these cells: were they experimentally characterized and used in the study, or are they only described as a conceptual starting point for the subsequent in silico analyses?
Reply: We thank the reviewer for this important observation. We have revised the manuscript to clarify the rationale for isolating SSC-derived ES-like cells. In this study, these cells were experimentally generated and characterized in our laboratory as a biological reference point to validate pluripotency potential through teratoma and chimera assays. However, the primary results presented in this manuscript focus on in silico transcriptomic analyses (GEO and TCGA datasets) to explore gene expression programs relevant to SSC biology and male infertility. Thus, the SSC-derived ES-like cells serve as an experimental proof-of-concept and biological context for our computational analyses, rather than forming the main dataset for the bioinformatics work. We have emphasized this distinction in the Introduction and Methods to avoid ambiguity.

Reviewer 2 Report
Comments and Suggestions for Authors
Comments and Suggestions:
Title: Identification of regulatory RNA binding genes in Spermatogonial Stem Cell Reprogramming to ES-like Cells Using Machine Learning–Integrated Transcriptomic and Network Analysis
Reviewer’s report:
The manuscript by Abroudi et al., presented an interesting study that demonstrated that Spermatogonial stem cells (SSCs) can be reprogrammed into ES-like cells. The authors have used mice model, transcriptomics, GEO datasets, WGCNA analysis to provide sufficient evidences. They confirmed that ES-like cells have strong expression of OCT4, DAZL, and VASA genes and WGCNA analysis identified key co-expression modules and hub regulatory RNA binding genes (Ctdsp1, Rest, and Stra8) potentially responsible for this reprogramming process which can be a potential application in male fertility preservation and stem cell-based therapies.
The manuscript provides novel approaches but some points need to be addressed.
- Please check sections 2.9 and 2.10 for text duplicacy.
- The results of WCGNA analysis are missing. Please add required figures which shows scale independence, mean connectivity, dendrogram, module trait relationships and also which modules were selected for further analysis.
- Line 337 and 338: Please check the number ‘2.3’? Also, the cutoff for log2 fold change of -2 and 2 was mentioned in methodology. What was the reason of selecting − 1.5 > fold change > 1.5?
- Figure 3: Please check the legends as they do not match with the sub-figures.
- How the two datasets GSE43850 and GSE38776 were analyzed? Please add how many samples from each dataset was used? It seems the two datasets have different platforms? How they can be directly used for DEG analysis?
- The terminology like SSCs, ES-like cells, and ESCs should be consistent in every place in the manuscript.
- Figure 4: The figure text is difficult to read. Please make it large.
Author Response
Reviewer’s report:
The manuscript by Abroudi et al., presented an interesting study that demonstrated that Spermatogonial stem cells (SSCs) can be reprogrammed into ES-like cells. The authors have used mice model, transcriptomics, GEO datasets, WGCNA analysis to provide sufficient evidences. They confirmed that ES-like cells have strong expression of OCT4, DAZL, and VASA genes and WGCNA analysis identified key co-expression modules and hub regulatory RNA binding genes (Ctdsp1, Rest, and Stra8) potentially responsible for this reprogramming process which can be a potential application in male fertility preservation and stem cell-based therapies.
The manuscript provides novel approaches but some points need to be addressed.
- Please check sections 2.9 and 2.10 for text duplicacy.
Reply: We thank the reviewer for this helpful observation. Upon careful review, we found that Sections 2.9 and 2.10 indeed contained duplicated methodological text. To address this, we have revised Section 2.10 to avoid redundancy. The updated version now refers back to the detailed methodology described in Section 2.9 and focuses specifically on the validation of hub genes with scRNA-seq datasets. This change improves the clarity and flow of the manuscript while eliminating unnecessary repetition.
- The results of WCGNA analysis are missing. Please add required figures which shows scale independence, mean connectivity, dendrogram, module trait relationships and also which modules were selected for further analysis.
Reply: We thank the reviewer for this important suggestion. We have revised the Results section to include the missing outputs of the WGCNA analysis (Section 3.6).
- Line 337 and 338: Please check the number ‘2.3’? Also, the cutoff for log2 fold change of -2 and 2 was mentioned in methodology. What was the reason of selecting − 1.5 > fold change > 1.5?
Reply: We thank the reviewer for pointing this out. The number “2.3” at line 337–338 was a typographical error, and it has now been corrected to “2,145,” which reflects the correct number of differentially expressed genes (DEGs) identified.
Regarding the fold-change cutoff, we agree that consistency is important. In the Materials and Methods section, we initially set the threshold at log2 fold change ≥ |2| for identifying significant DEGs. However, during comparative analysis between ESCs and SSCs, we also reported results using a more relaxed cutoff of log2 fold change ≥ |1.5| to capture additional biologically relevant genes that were not detected at the stricter threshold. This dual approach allowed us to balance stringency (high-confidence DEGs at |2| cutoff) with sensitivity (inclusion of additional candidates at |1.5| cutoff), as has been used in prior transcriptomic studies.
To avoid confusion, we have revised the text for clarity and explicitly explained why both thresholds were used. The Results section now reads:
“Out of 2,145 DEGs, we applied a primary cutoff of log2 fold change ≥ |2| for stringent DEG selection. In addition, to capture a broader set of biologically relevant genes, a secondary analysis was performed at log2 fold change ≥ |2|, which identified additional candidates for downstream validation.”
- Figure 3: Please check the legends as they do not match with the sub-figures.
Reply: We corrected in Figure 3.
- How the two datasets GSE43850 and GSE38776 were analyzed? Please add how many samples from each dataset was used? It seems the two datasets have different platforms? How they can be directly used for DEG analysis?
Reply: We appreciate the reviewer’s careful observation. In the revised manuscript, we have clarified the dataset selection and analysis process:
From GSE43850, we included 3 ESC samples and 3 ES-like cell samples.
From GSE38776, we included 3 SSC samples.
It is correct that the two datasets were generated using different array platforms. To ensure comparability, all raw CEL files were reprocessed using the Robust Multi-array Average (RMA) normalization method within the Transcriptome Analysis Console (TAC v4.0). Batch effects between platforms were corrected using the ComBat function from the sva package in R, which adjusts for technical differences while preserving biological variation.
After normalization and batch correction, the combined expression matrix was used for differential expression analysis with the empirical Bayes (eBayes) ANOVA method. This allowed us to directly compare SSCs, ES-like cells, and ESCs across datasets.
We have now revised the Materials and Methods (Section 2.6) to include this information, and we have updated the Results section to specify the sample numbers and preprocessing workflow.
- The terminology like SSCs, ES-like cells, and ESCs should be consistent in every place in the manuscript.
Reply: We thank the reviewer for this important suggestion. We have carefully revised the manuscript to ensure consistency in terminology throughout. Specifically:
Spermatogonial stem cells are now uniformly referred to as SSCs.
Embryonic stem cell-like cells are consistently referred to as ES-like cells.
Embryonic stem cells are consistently referred to as ESCs.
- Figure 4: The figure text is difficult to read. Please make it large.
Reply: Done.

Round 2
Reviewer 1 Report
Comments and Suggestions for Authors
The authors addressed all my observations, and I recommend the publication of this manuscript.